# De novo fatty-acid synthesis protects invariant NKT cells from cell death, thereby promoting their homeostasis and pathogenic roles in airway hyperresponsiveness

Jaemoon Koh[1,2†], Yeon Duk Woo[2†], Hyun Jung Yoo[3], Jun-Pyo Choi[4], Sae Hoon Kim[4,5], Yoon-Seok Chang[4,5], Kyeong Cheon Jung[1], Ji Hyung Kim[3], Yoon Kyung Jeon[1], Hye Young Kim[2], Doo Hyun Chung[1,2]*

[1]Department of Pathology, Seoul National University College of Medicine, Seoul, Republic of Korea; [2]Laboratory of Immune Regulation in Department of Biomedical Sciences, Seoul National University College of Medicine, Seoul, Republic of Korea; [3]Laboratory of Immunology and Vaccine Innovation, Department of Biotechnology, College of Life Sciences and Biotechnology, Korea University, Seoul, Republic of Korea; [4]Department of Internal Medicine, Seoul National University Bundang Hospital, Seongnam, Republic of Korea; [5]Institute of Allergy and Clinical Immunology, Seoul National University Medical Research Council, Seoul, Republic of Korea

*For correspondence:
doohyun@snu.ac.kr

†These authors contributed equally to this work

Competing interest: The authors declare that no competing interests exist.

**Abstract** Invariant natural-killer T (*i*NKT) cells play pathogenic roles in allergic asthma in murine models and possibly also humans. While many studies show that the development and functions of innate and adaptive immune cells depend on their metabolic state, the evidence for this in *i*NKT cells is very limited. It is also not clear whether such metabolic regulation of *i*NKT cells could participate in their pathogenic activities in asthma. Here, we showed that acetyl-coA-carboxylase 1 (ACC1)-mediated de novo fatty-acid synthesis is required for the survival of *i*NKT cells and their deleterious functions in allergic asthma. ACC1, which is a key fatty-acid synthesis enzyme, was highly expressed by lung *i*NKT cells from WT mice that were developing asthma. *Cd4*-Cre::*Acc1*[fl/fl] mice failed to develop OVA-induced and HDM-induced asthma. Moreover, *i*NKT cell-deficient mice that were reconstituted with ACC1-deficient *i*NKT cells failed to develop asthma, unlike when WT *i*NKT cells were transferred. ACC1 deficiency in *i*NKT cells associated with reduced expression of fatty acid-binding proteins (FABPs) and peroxisome proliferator-activated receptor (PPAR)γ, but increased glycolytic capacity that promoted *i*NKT-cell death. Furthermore, circulating *i*NKT cells from allergic-asthma patients expressed higher *ACC1* and *PPARG* levels than the corresponding cells from non-allergic-asthma patients and healthy individuals. Thus, de novo fatty-acid synthesis prevents *i*NKT-cell death via an ACC1-FABP-PPARγ axis, which contributes to their homeostasis and their pathogenic roles in allergic asthma.

## eLife assessment

The study's results offer a **fundamental** insight into how ACC1-mediated fatty-acid synthesis affects the survival and pathogenicity of iNKT cells in allergic asthma. The inclusion of mouse models, involving genetic adjustments and reconstitution experiments, along with the disparities found in

iNKT cells between allergic asthma patients and control subjects in human studies, adds **compelling** evidence that substantiates these findings.

## Introduction

Invariant natural-killer T (iNKT) cells are a subset of innate T cells that express a restricted T-cell receptor (TCR) repertoire (Vα24-Jα18 in humans and Vα14-Jα18 in mice) and respond to glycolipids presented on the CD1d molecule. One such glycolipid is α-galactosylceramide (α-GalCer), which originates from a marine sponge and strongly activates iNKT cells in vitro and in vivo (*Allende et al., 2008*; *Brossay et al., 1998*). Once activated, iNKT cells secrete a diverse array of cytokines, including interferon (IFN)γ, interleukin (IL)-4, IL-13, IL-17A, and IL-10 (*Abdollahi-Roodsaz et al., 2008*). These iNKT cell-derived cytokines play important roles in the pathogenesis of various immunological diseases, including sepsis, arthritis, tumors, and asthma (*Osuchowski et al., 2018*; *Ma et al., 2018*; *Terabe and Berzofsky, 2018*; *Kim et al., 2005*). With regard to the latter, iNKT cells are one of the key effectors that promote allergic asthma in murine models. Specifically, by secreting IL-4, iNKT cells stimulate the differentiation of T-helper (Th)2 cells and their responses. Moreover, by secreting IL-13, iNKT cells promote the contraction of airway smooth muscle, which is a cardinal feature of asthma (*Akbari et al., 2003*; *Manson et al., 2020*). In addition, we recently reported that iNKT cells contribute to the development of asthma by secreting X-C motif chemokine ligand-1, which recruits conventional X-C motif chemokine receptor 1-expressing type-1 dendritic cells into the lungs: this then stimulates the Th2 responses that drive asthma (*Woo et al., 2018*). Thus, iNKT cell-derived soluble factors play multiple critical roles in the development of allergic asthma in murine models. It should be noted that it is somewhat unclear what role iNKT cells play in human asthma. Several studies suggest that human asthmatic airways contain large numbers of iNKT cells, but this was not observed by other studies (*Akbari et al., 2006*; *Das et al., 2006*; *Foell et al., 2007*; *Hamzaoui et al., 2006*; *Pham-Thi et al., 2006*; *Thomas et al., 2006*). However, a recent review describes additional lines of evidence that suggest that iNKT cells do contribute to the development and exacerbation of human asthma development, although the mechanisms may not necessarily involve increased iNKT cell frequencies. Thus, further research on the roles of iNKT cells in mouse models and humans with asthma is needed to improve our understanding of the pathogenesis of this disease.

Many studies show that the development and functions of innate and adaptive immune cells are dependent on the metabolic states of these cells (*Jung et al., 2019*). In particular, the metabolic balance between glycolysis and oxidative phosphorylation (OXPHOS) shapes the differentiation and functions of CD4$^+$ Th cells, Foxp3$^+$ regulatory T cells (Tregs), cytotoxic CD8$^+$ T cells, and macrophages (*Ganeshan and Chawla, 2014*). The development and functions of iNKT cells may also depend on their metabolism: while the evidence to date is quite limited (*Yarosz et al., 2021*), the thymic development of iNKT cells and their inflammatory cytokine production as mature cells associate with changes in their glycolysis:OXPHOS balance (*Kim et al., 2017*). Moreover, peripheral iNKT-cell functions are regulated by their levels of reactive oxygen species, which are by-products of aerobic metabolism (*Kumar et al., 2019*). It is also possible that immune-cell metabolism may participate in asthma since a recent study showed that fatty acid-oxidation inhibitors alleviate house-dust mite (HDM)-induced airway inflammation (*Al-Khami et al., 2017*). Similarly, the airway Th2 cells that mediate allergic responses against HDM depend on glycolysis and lipid metabolism (*Tibbitt et al., 2019*).

Several studies show that lipid biogenesis, which is a major metabolic pathway, also affects the differentiation and functions of conventional CD4$^+$ T cells (*Berod et al., 2014*; *Soroosh et al., 2014*; *Wang et al., 2015*): for example, a CD5-like molecule enforces low cholesterol biosynthesis in Th17 cells, which limits ligand availability for Rorγt, the Th17 master transcription factor, thereby preventing Th17 cells from becoming pathogenic and inducing autoimmunity (*Tibbitt et al., 2019*). Similarly, several studies show that acetyl-CoA-carboxylase 1 (ACC1) affects multiple T-cell subsets. ACC1 is a key fatty-acid synthesis enzyme in the cytosol that catabolizes the ATP-dependent carboxylation of acetyl-CoA to malonyl-CoA (*Wakil and Abu-Elheiga, 2009*), and its downregulation impairs the peripheral persistence and homeostatic proliferation of CD8$^+$ T cells, augments Treg development from naïve CD4$^+$ T cells and their suppressive functions, blocks the development of Th17 cells from naïve CD4$^+$ T cells (*Berod et al., 2014*; *Huang et al., 2014*), impairs the formation of IL-5-producing CD4$^+$ T cells (*Nakajima et al., 2021*), and promotes CD4$^+$ T cell development (*Endo et al., 2019*).

The T-cell changes caused by inhibiting ACC1 reduce disease severity in murine models of listeria infection, chronic graft-versus-host disease, multiple sclerosis (*Lee et al., 2014*), and T cell-mediated asthma (*Nakajima et al., 2021*); it also promotes resistance to parasites (*Endo et al., 2019*). Thus, by shaping fatty-acid synthesis, ACC1 can alter pathogenic and protective T-cell responses.

To our knowledge, the role(s) of ACC1-mediated fatty-acid synthesis in other immune cells, including *i*NKT cells, and in asthma have not been assessed. In this study, we used ACC1-deficient (*Cd4*-Cre::*Acc1*fl/fl) mice and adoptive transfer experiments to determine whether ACC1-mediated fatty-acid metabolism in *i*NKT cells participates in the development of asthma. Indeed, we found that ACC1 promotes the survival of *i*NKT cells by regulating a fatty acid-binding protein (FABP)-PPARγ axis that can downregulate glycolysis. The improved ACC1-mediated *i*NKT-cell survival shapes the homeostasis and functions of these cells, which in turn promote the development of ovalbumin (OVA)- or HDM-induced allergic asthma in mice. We also observed that the ACC1-FABP-PPARγ axis may be active in patients with allergic asthma but not controls.

## Results

### Activated *i*NKT cells express high levels of a fatty-acid synthesis-related gene whose deletion blocks allergen-induced asthma

To determine whether intracellular metabolic processes in *i*NKT cells play important roles in allergic asthma, *i*NKT cells sorted from the lungs of OVA-induced asthma model mice were assessed for the expression of key enzymes involved in glycolysis, the tricarboxylic acid (TCA) cycle, β-oxidation, and fatty-acid synthesis (*Figure 1*). Indeed, OVA challenge significantly increased *i*NKT cell expression of β-oxidation and fatty-acid synthesis enzymes and tended to elevate TCA-cycle enzyme expression. However, glycolysis-enzyme expression remained low (*Figure 1A, B*). The significantly upregulated fatty-acid synthesis enzymes were Acc1, Fasn, and Oxsm: Acc1 and Fasn play a particularly key role in de novo fatty-acid synthesis. Notably, when purified murine *i*NKT cells and conventional CD4+ and CD8+ T cells were activated in vitro with anti-CD3/CD28, the *i*NKT cells expressed much higher levels of *Acc1* and *Fasn* than the other cells. By contrast, the CD4+ and CD8+ T cells expressed higher levels of TCA cycle (*Sdh2b* and *Idh1a*) and glycolysis (*Hk2* and *Pkm2*) enzymes, respectively (*Figure 1C*). This is consistent with the previous gene set enrichment analysis of Oh et al.: when they compared unstimulated murine *i*NKT and CD4+ T cells to each other, they found that genes involved in the 'lipid, fatty acid and steroid metabolism' pathway were enriched in *i*NKT cells (*Figure 1—figure supplement 1A*; *Oh et al., 2011*). Thus, *i*NKT cells express *Acc1* and *Fasn* when they are stimulated, including in asthma, whereas conventional CD4+ and CD8+ T cells do not.

We then focused on ACC1. To test whether ACC1-mediated de novo fatty-acid synthesis in *i*NKT cells shapes AHR and allergic immune responses, we generated *Cd4*-Cre::*Acc1*fl/fl mice: the CD4-positive cells (i.e. conventional CD4+ T and *i*NKT cells) in these mice cannot express ACC1. Since conventional CD4+ T cells did not upregulate ACC1 when stimulated in vitro or in vivo (*Figure 1B and C*), the knockout is likely to be physiologically relevant for *i*NKT cells only. When the *Cd4*-Cre::*Acc1*fl/fl and *Acc1*fl/fl control mice were induced to undergo OVA-mediated asthma, we indeed observed that the loss of ACC1 expression attenuated AHR (*Figure 1D*) and the total immune-cell numbers in the lung and the bronchoalveolar lavage fluid (BALF) (*Figure 1—figure supplement 1B*). These changes associated with significantly decreased total lymphocyte, neutrophil, eosinophil, and macrophage frequencies in the BALF (*Figure 1—figure supplement 1C*) and lower eosinophil frequencies in the lung (*Figure 1—figure supplement 1D*). The lung CD8+ T cell and alveolar- and interstitial-macrophage frequencies did not change. Importantly, the total lung CD4+ T cell population (excluding *i*NKT cells) also did not change significantly (*Figure 1—figure supplement 1D*). By contrast, there was a sharp drop in *i*NKT cells in the lung; this was also associated with a significant increase in PI staining (*Figure 1E*). qRT-PCR analysis of the conventional lung CD4+ T cell population (excluding *i*NKT cells) showed that loss of ACC1 expression associated with lower *Gata3*, *Il4*, *Il5*, and *Il13* expression but similar *Tbx21*, *Rorc*, *Foxp3*, *Ifng*, *Il17a*, and *Il10* expression (*Figure 1—figure supplement 1E, F*). Indeed, when the lung *i*NKT cells and conventional CD4+ T cells were separately isolated, both *i*NKT cells and the conventional CD4+ T cells demonstrated dramatically lower *Il4* and *Il13* expression but similar *Ifng* and *Il17a* expression (*Figure 1F*, *Figure 1—figure supplement 1F*). Notably, all of these findings were recapitulated when we conducted the same analyses in mice that underwent

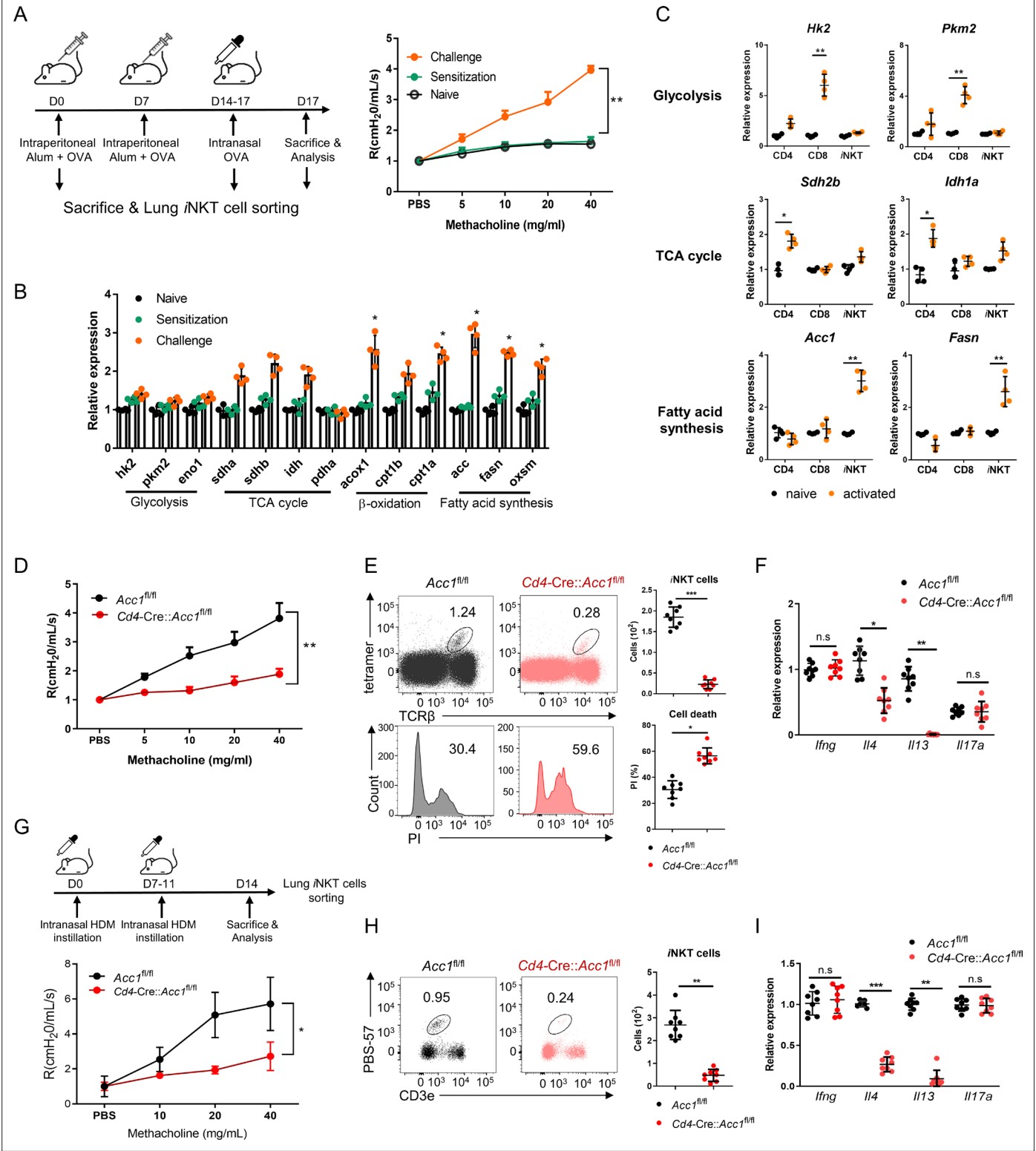

**Figure 1.** The fatty-acid synthesis pathway is upregulated in invariant natural-killer T (*i*NKT) cells during murine allergic asthma. (**A, B**) WT mice were subjected to ovalbumin (OVA)-induced asthma. (**A**) Schematic depiction of the OVA-induced allergic asthma model (left). The airway resistance of naïve, OVA-sensitized, and OVA-sensitized and OVA-challenged WT mice (right). (**B**) Gene expression of enzymes related to glycolysis, the tricarboxylic acid (TCA) cycle, β-oxidation, and fatty-acid synthesis in lung *i*NKT cells that were sorted on d0, d7, and d17. The data shown in (**A**) and (**B**) are pooled data from four independent experiments (n = 8/group). (**C**) In vitro gene expression of enzymes related to glycolysis (*Hk2* and *Pkm2*), the TCA cycle (*Sdh2b*

*Figure 1 continued on next page*

*Figure 1 continued*

and *Idh1a*), and fatty-acid synthesis (*Acc1* and *Fasn*) in CD8$^+$ T, CD4$^+$ T, and *i*NKT cells before and after CD3/CD28 stimulation. Representative results from two independent experiments (n = 4/group) are shown. (**D–I**) *Cd4*-Cre*Accl1*$^{fl/fl}$ mice and the *Acc1*$^{fl/fl}$ control mice were subjected to OVA-induced asthma (**D–F**) or house-dust mite (HDM)-induced asthma (**G–I**). (**D, G**) Airway resistance. (**E, H**) Numbers and survival status of *i*NKT cells that were sorted from the lung by flow cytometry. (**F, I**) Cytokine expression in *i*NKT cells sorted from the lung. The data shown in (**D–I**) are pooled data from four independent experiments (n = 8/group). All data are presented as mean ± SEM. ns, not significant. *p<0.05, **p<0.01, ***p<0.001, as determined by unpaired two-tailed *t*-tests.

The online version of this article includes the following figure supplement(s) for figure 1:

**Figure supplement 1.** *Cd4*-Cre::*Acc1*$^{fl/fl–}$ mice fail to develop airway resistance in the house-dust mite (HDM)-induced asthma model.

HDM-induced asthma (*Figure 1G–I* and *Figure 1—figure supplement 1G–K*). Thus, the inability of lung *i*NKT cells to express ACC1 induced *i*NKT-cell death during allergen-induced asthmogenesis. This also reduced their expression of *Gata3*, *Il4*, *Il5*, and *Il13* in lung conventional CD4$^+$ T cells: these effects markedly reduced the eosinophilia and AHR. These effects were independent of the inability of conventional lung CD4+ T cells to express ACC1 in the mice.

## ACC1-deficient *i*NKT cells cannot generate α-GalCer-mediated AHR and their adoptive transfer into *i*NKT cell-deficient mice cannot restore allergen-induced asthma

The experiments above suggest that *i*NKT cells alone mediate ACC1-mediated regulation of airway inflammation. To test this further, we challenged *Cd4*-Cre::*Acc1*$^{fl/fl}$ and *Acc1*$^{fl/fl}$ mice intranasally with α-GalCer, which has been shown previously to induce *i*NKT cell activation-mediated AHR (*Meyer et al., 2006*). Consistent with the results from OVA- and HDM-induced asthma models, the *Cd4*-Cre::*Acc1*$^{fl/fl}$ mice showed the following: less AHR and lung and BALF inflammation; significantly fewer lymphocytes, neutrophils, and macrophages in the BALF; less eosinophilia in the BALF and lungs; no difference in lung CD4$^+$ and CD8$^+$ T cell frequencies but significantly fewer lung *i*NKT cells and greater PI staining of these cells; lower expression by the lung conventional CD4$^+$ T cell population of *Gata3*, *Il4*, *Il5*, and *Il13* but not *Tbx21*, *Rorc*, *Foxp3*, *Ifng*, *Il17a*, or *Il10*; and similar *Ifng* and *Il17a* expression but dramatically reduced *Il4* and *Il13* expression by the lung *i*NKT cells (*Figure 2A–H*). Thus, ACC1-deficient *i*NKT cells also failed to promote α-GalCer-induced AHR.

To further confirm that it is the ACC1 in *i*NKT cells that regulated AHR, we adoptively transferred an equal number of WT *i*NKT cells (i.e. from *Acc1*$^{fl/fl}$ mice) or ACC1-deficient *i*NKT cells into Jα18 KO mice, which lack NKT cells (*Chandra et al., 2015*). The mice were then subjected to OVA-induced asthma. The WT *i*NKT cells, but not the ACC1-deficient *i*NKT cells, restored the following features: AHR (*Figure 3A*); the high total immune-cell numbers in the BALF and lungs (*Figure 3B*); the high lymphocyte, neutrophil, eosinophil, and macrophage frequencies in the BALF and lungs (*Figure 3C and D*); the high *Gata3*, *Il4*, *Il5*, and *Il13* expression by the lung CD4$^+$ T cell population (*Figure 3E and F*). As expected, neither adoptively transferred cell type affected the *Tbx21*, *Rorc*, *Foxp3*, *Ifng*, *Il17a*, and *Il10* expression by the lung conventional CD4$^+$ T cell population (*Figure 3E and F*). Moreover, the transfer of WT *i*NKT cells restored the *i*NKT cell numbers in the lungs and their expression of *Il4* and *Il13* without changing their *Ifng* and *Il17a* expression; by contrast, transfer of ACC1-deficient *i*NKT cells led to very low *i*NKT-cell frequencies in the lung and low expression of all cytokines (*Figure 3G and H*). In addition, the adoptively transferred ACC1-deficient *i*NKT cells in the lung displayed more apoptosis than the transferred WT *i*NKT cells (*Figure 3I*). Lastly, transfer of the WT *i*NKT cells associated with significantly higher histological lung scores than transfer of the ACC1-deficient *i*NKT cells (*Figure 3J*). Thus, ACC1-deficient *i*NKT cells cannot promote AHR in allergic asthma models.

## ACC1-deficient *i*NKT cells exhibit defects in thymic development, homeostasis, and activation

To investigate the effects of ACC1-mediated metabolic programming on the development and function of *i*NKT cells, we examined the thymic development and peripheral homeostasis of *i*NKT cells in *Cd4*-Cre::*Acc1*$^{fl/fl}$ mice. Mature *Cd4*-Cre::*Acc1*$^{fl/fl}$ mice had fewer *i*NKT cells in the thymus, spleen, and liver than *Acc1*$^{fl/fl}$ mice (*Figure 3—figure supplement 1A*). By contrast, the conventional CD4$^+$ and CD8$^+$ T cell numbers in the thymus and spleen were not affected by *Cd4*-Cre::*Acc1*$^{fl/fl}$ (*Figure 3—figure*

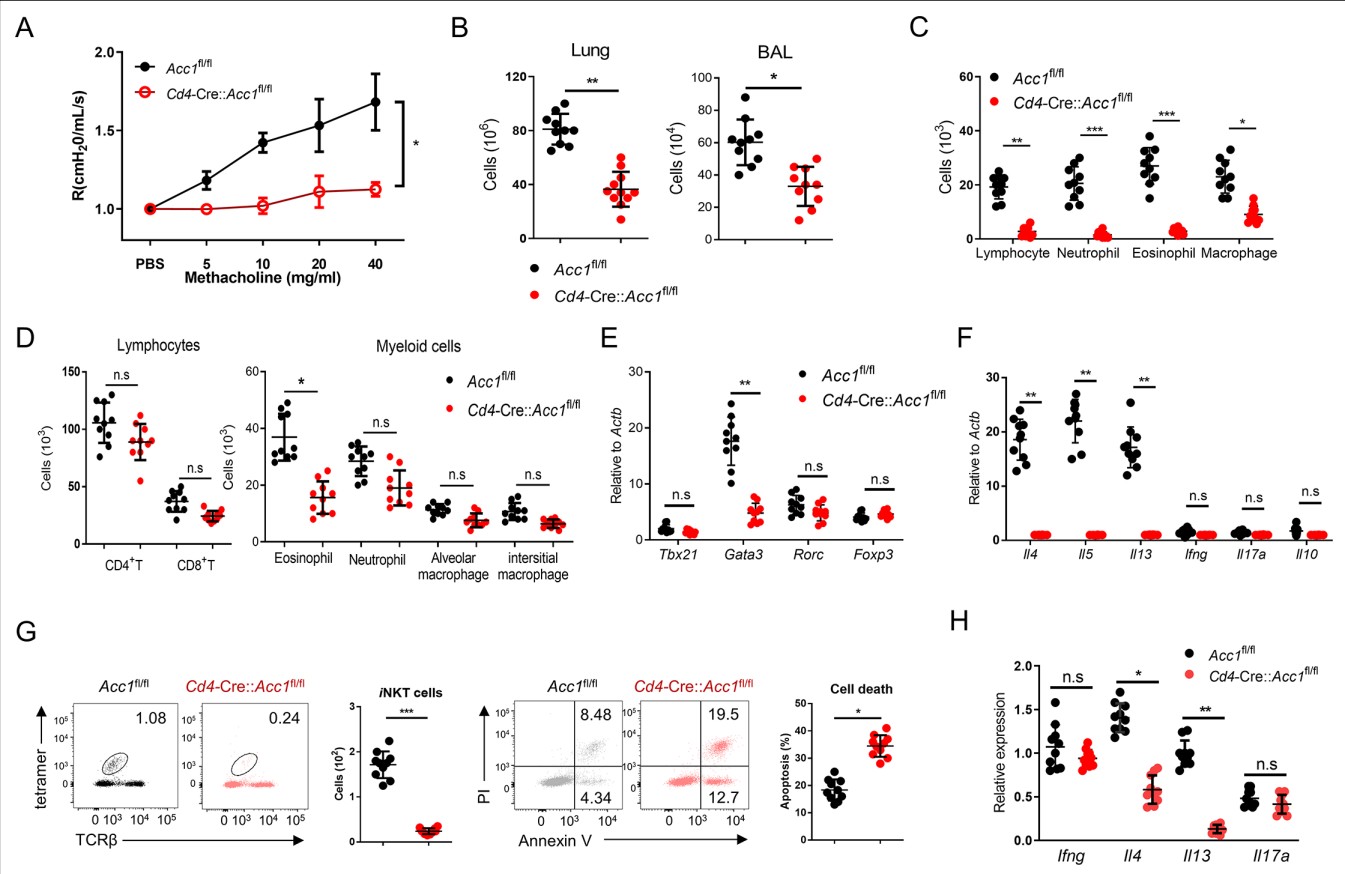

**Figure 2.** ACC1 deficiency in murine invariant natural-killer T (*i*NKT) cells associates with failure to develop α-GalCer-induced AHR. *Cd4*-Cre::*Acc1*fl/fl and *Acc1*fl/fl mice were administered α-GalCer intranasally. (**A**) Airway resistance. (**B**) Total immune cell numbers in the lungs and bronchoalveolar lavage fluid (BALF). (**C, D**) Numbers of immune-cell subsets in the BALF (**C**) and lungs (**D**). (**E, F**) Expression levels of transcription factors (**E**) and cytokines (**F**) in the conventional CD4⁺ T cell population that was sorted from the lungs. (**G**) Numbers and survival status of *i*NKT cells sorted from the lungs. (**H**) Cytokine expression in *i*NKT cells sorted from the lungs. All data are pooled from five independent experiments (n = 10/group) and presented as mean ± SEM. ns, not significant. *p<0.05, **p<0.01, ***p<0.001, as determined by unpaired two-tailed *t*-tests.

*supplement 1B*). However, the ACC1 deletion did not alter the frequencies of the NKT1, NKT2, and NKT17 subsets in the *i*NKT cell population at any of the developmental stages, namely, stage S1 (CD24⁺c-Kit⁻CD44⁻), stage S2 (CD24⁺c-Kit⁻CD44⁺), and stage S3 (CD24⁺c-Kit⁺CD44⁺) (*Figure 3—figure supplement 1C*). Thus, ACC1 deficiency appeared to reduce *i*NKT-cell numbers.

This was tested further by generating mixed bone marrow (BM) chimera mice: thus, a 1:1 ratio of CD45.1⁺ WT BM cells and CD45.2⁺ *Cd4*Cre::*Acc1*fl/fl BM cells were transferred into lethally irradiated CD45.2⁺ WT recipient mice (*Figure 3—figure supplement 1D*). The ACC1-deficient BM cells generated markedly fewer *i*NKT cells in the thymus, spleen, and liver than the WT BM cells (*Figure 3—figure supplement 1E, F*). This confirmed that ACC1 deficiency induced an intrinsic defect in *i*NKT-cell thymic development and peripheral homeostasis. Moreover, when *i*NKT cells sorted from the livers of *Cd4*-Cre::*Acc1*fl/fl and *Acc1*fl/fl mice underwent TCR ligation with anti-CD3/CD28 in vitro, the ACC1-deficient cells produced less CD69 activation marker and cytokine (*Figure 3—figure supplement 1G, H*).

Thus, ACC1 plays a cell-intrinsic role in the regulation of *i*NKT-cell thymic development, peripheral homeostasis, and activation.

## Enhanced glycolysis promotes the cell death of ACC1-deficient *i*NKT cells

To investigate the mechanism(s) by which ACC1-mediated metabolic programming of *i*NKT cells promotes allergic asthma, we subjected the sorted unstimulated *i*NKT cells from *Cd4*-Cre::*Acc1*fl/fl and

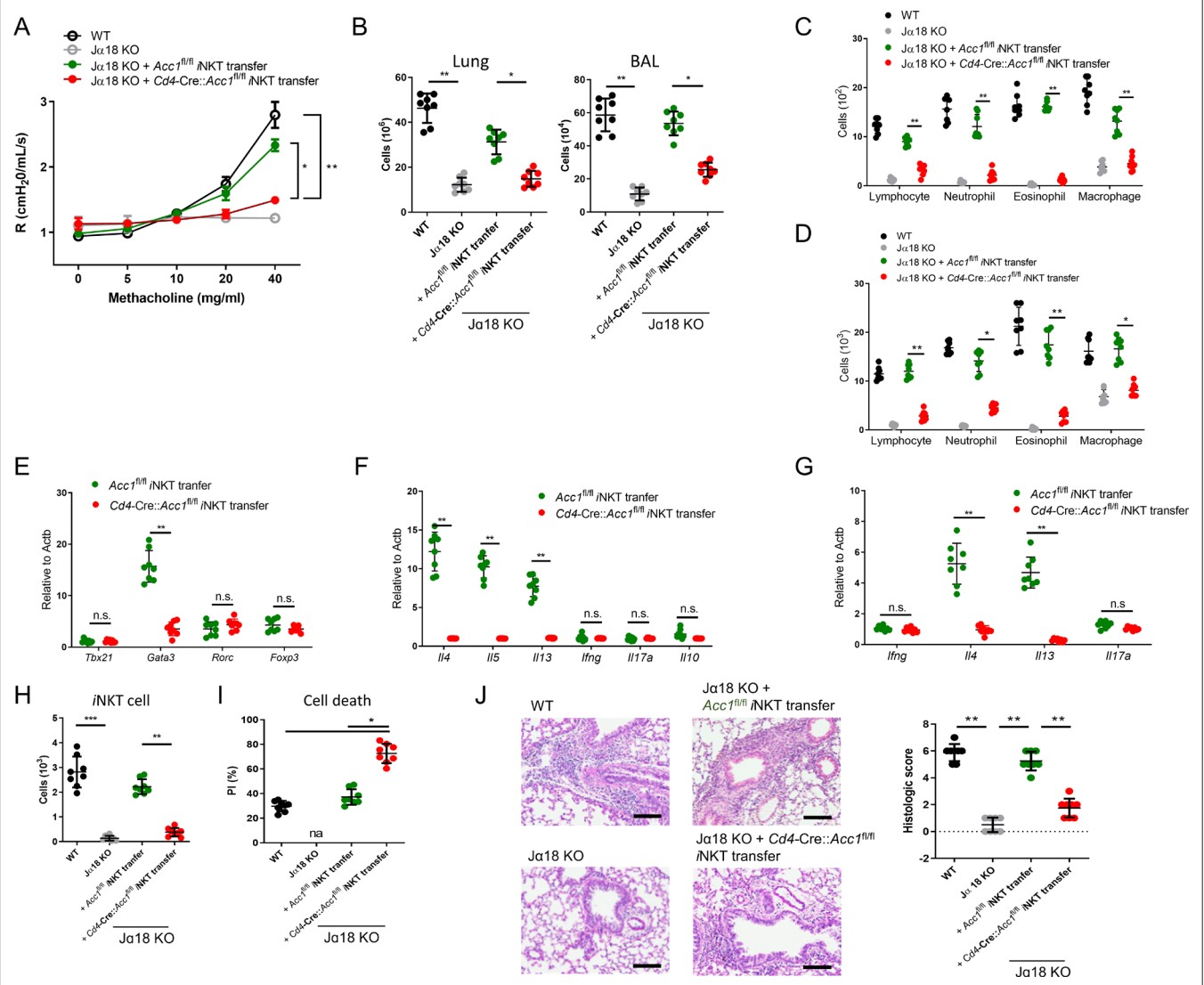

**Figure 3.** Adoptive transfer of WT, but not ACC1-deficient, invariant natural-killer T (iNKT) cells restores the ability of NKT-knockout mice to develop ovalbumin (OVA)-induced asthma. iNKT cells sorted from naïve Cd4-Cre::Acc1^fl/fl and Acc1^fl/fl mice were adoptively transferred into naïve WT and Jα18 KO mice, after which OVA-induced asthma was initiated. (**A**) Airway resistance. (**B**) Total inflammatory cell numbers in the lungs and bronchoalveolar lavage fluid (BALF). (**C, D**) Numbers of immune-cell subsets in BALF (**C**) and lungs (**D**). (**E, F**) Expression levels of transcription factors (**E**) and cytokines (**F**) in CD4+ T cells sorted from the lungs. (**G**) Cytokine expression in iNKT cells sorted from the lungs. (**H, I**) Numbers (**H**) and survival status (**I**) of iNKT cells from the lungs. (**J**) Representative histological lung section images and histological scores. Scale bar = 200 µM. The data in (**A–J**) are pooled data from four independent experiments (n = 8/group). All data are presented as mean ± SEM. ns, not significant. *p<0.05, **p<0.01, ***p<0.001, as determined by unpaired two-tailed t-tests.

The online version of this article includes the following figure supplement(s) for figure 3:

**Figure supplement 1.** Deficiency of ACC1 in invariant natural-killer T (iNKT) cells perturbed their thymic development and peripheral homeostasis in cell intrinsic manner.

Acc1^fl/fl mice to transcriptome and metabolome analyses (**Figure 4A–D**). The ACC1-deficient iNKT cells displayed distinct metabolic reprogramming compared to WT iNKT cells. Specifically, they exhibited low fatty-acid synthesis but high glycolysis. Indeed, compared to WT iNKT cells, their extracellular acidification rate (ECAR), the level of 2-NBDG, which is an indicator of glucose uptake, and expression of Glut1 and other glycolysis-related genes were higher while their expression of gluconeogenesis genes were lower. However, their oxygen consumption rate (OCR) was similar (**Figure 4E–H**). Furthermore, the transcriptome analysis revealed significantly altered levels of cell death-related genes in ACC1-deficient iNKT cells (**Figure 4C**). These observations suggest that ACC1-deficient iNKT cells

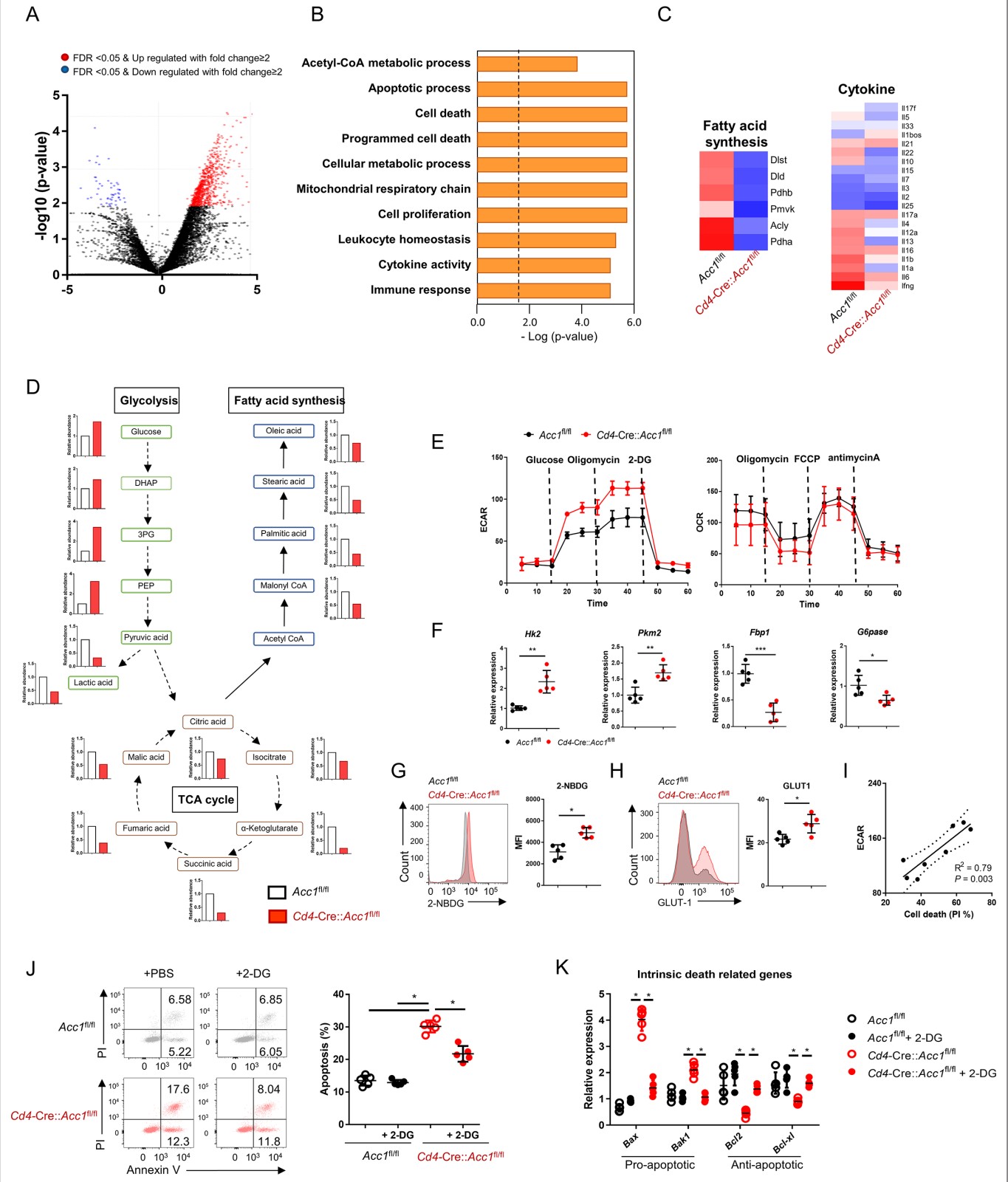

**Figure 4.** The high glycolytic capacity of ACC1-deficient invariant natural-killer T (*i*NKT) cells associates with their enhanced cell death. *i*NKT cells were sorted from the liver of naïve *Cd4*-Cre::*Acc1*fl/fl and *Acc1*fl/fl mice and used directly in the experiments. The two *i*NKT cell preparations were compared. (**A**) Volcano plot of the gene expression. (**B**) Signature gene expression, as determined by gene set enrichment analysis (GSEA). (**C**) Heat map of the fatty-acid synthesis, glycolysis, cell death, and cytokine genes. (**D**) Metabolome data map of the glycolysis, tricarboxylic acid (TCA) cycle,

*Figure 4 continued on next page*

*Figure 4 continued*

and fatty-acid synthesis metabolites. (**E**) Seahorse extracellular flux analysis of extracellular acidification rate (ECAR) and oxygen consumption rate (OCR). (**F**) Expression of glycolysis-related (*Hk2*, *Pkm2*) and gluconeogenesis-related (*Fbp1*, *G6pase*) genes. (**G**) 2-NBDG uptake. (**H**) GLUT1 expression. (**I**) Correlation between *i*NKT-cell death and maximal glycolytic activity. (**J, K**) The *i*NKT cells were treated with 2-DG to inhibit glycolysis. The effect of 2-DG on cell death (**J**) and intrinsic cell death-related gene expression (**K**) was then determined. In (A–D), the *i*NKT cells were from 8 mice/group. The data in (**D**) are presented as mean ± SEM. The data in (**E–K**) are pooled data from five independent experiments (n = 5/group) and are presented as mean ± SEM. ns, not significant. *p<0.05, **p<0.01, ***p<0.001, as determined by unpaired two-tailed *t*-tests.

are prone to dying and that this may correlate positively with glycolysis. Indeed, the ECAR and cell death in ACC1-deficient *i*NKT cells correlated positively (*Figure 4I*). This relationship was confirmed by treating the ACC1-deficient *i*NKT cells with 2-deoxy-D-glucose (2-DG), which inhibits glycolysis (*Pajak et al., 2019*): this decreased their cell death and expression of the pro-apoptotic genes while increasing their expression levels of anti-apoptotic genes. These effects were not observed in WT *i*NKT cells (*Figure 4J and K*). Thus, the high levels of glycolysis in ACC1-deficient *i*NKT cells promote their cell death.

## ACC1-mediated metabolic programming promotes PPARγ expression and the subsequent survival of *i*NKT cells

To identify the target genes that regulate the metabolic reprogramming and cell death in ACC1-deficient *i*NKT cells, we analyzed the transcription factors in the transcriptome data described above. This showed that sorted unstimulated ACC1-deficient *i*NKT cells expressed *Pparg* at significantly lower levels than WT *i*NKT cells (*Figure 5A and B*). This was also observed when the cells were stimulated with anti-CD3/CD28: the treatment elevated *Pparg* expression in both cell types but much less in the ACC1-deficient cells (*Figure 5B*). Thus, PPAR-γ may play a critical role in ACC1-mediated modification of *i*NKT-cell functions. This is consistent with the fact that PPAR-γ is important for activating genes involved in lipid biosynthesis and the maintenance of metabolic homeostasis (*Ahmadian et al., 2013*; *Grygiel-Górniak, 2014*). Indeed, when anti-CD3/CD28-stimulated ACC1-deficient *i*NKT cells were treated with the PPARγ agonist pioglitazone, their expression of *Pparg* was not only significantly restored, their expression of the gluconeogenesis genes *Cebpa*, *Fbp1*, and *G6pc1* rose while their 2-NBDG uptake, expression of Glut1 and the glycolysis genes *Hk2* and *Pkm2*, and cell death dropped (*Figure 5B–D*).

ACC1 catalyzes the carboxylation of acetyl-CoA to malonyl-CoA, which is then converted to the long-chain fatty acid palmitate (*Moon et al., 2001*). We found that treatment with palmitate had similar effects on CD3/CD28-stimulated ACC1-deficient *i*NKT cells as pioglitazone, namely, it increased *Pparg* and gluconeogenesis gene expression and downregulated glycolysis (*Figure 5E*). These effects were also observed when unstimulated ACC1-deficient *i*NKT cells were treated with palmitate: *Pparg* expression rose (*Figure 5G*) while glucose uptake and cell death dropped (*Figure 5H*). Thus, restoring the levels of palmitate, the downstream product of ACC1 activity, restored PPARγ expression, which in turn downregulated glycolysis and thereby protected the cells from cell death.

PPARγ activity can be regulated by a variety of ligands, including fatty acids, whose transport to the nucleus is actively facilitated by fatty acid-binding proteins (FABPs) (*Wolfrum et al., 2001*; *Patil et al., 2019*; *Hou et al., 2022*). The unstimulated ACC1-deficient *i*NKT cells showed depressed expression of the genes encoding FABP1, FABP3, and FABP5 (*Figure 5F*). Thus, ACC1 deficiency downregulated the expression of these genes. We then asked whether circumventing the ACC1 deficiency by treating the unstimulated cells with palmitate could improve the expression of these FABPs, as exogenous treatment of palmitate has been shown to induce FABPs in macrophages: indeed, *Fabp1*, *Fabp3*, and especially *Fabp5* expression were upregulated in *i*NKT cells as well (*Figure 5F*).

We then determined whether the FABPs mediate the proglycolytic and pro-apoptotic effects of ACC1 (and palmitate) deficiency in *i*NKT cells by culturing unstimulated palmitate-treated ACC1-deficient *i*NKT cells with a FABP inhibitor, FABP-IN-1, which is known to inhibit FABP 3, 5, and 7. Indeed, the inhibitor blocked the ability of palmitate to restore PPARγ expression (*Figure 5G*) and suppress glucose uptake and cell death (*Figure 5H*). These findings were also observed when CD3/CD28-stimulated ACC1-deficient *i*NKT cells were treated with palmitate with and without FABP inhibitor: the FABP inhibitor reversed the ability of palmitate to increase *Pparg* and gluconeogenesis gene expression and downregulate glycolysis (*Figure 5E*).

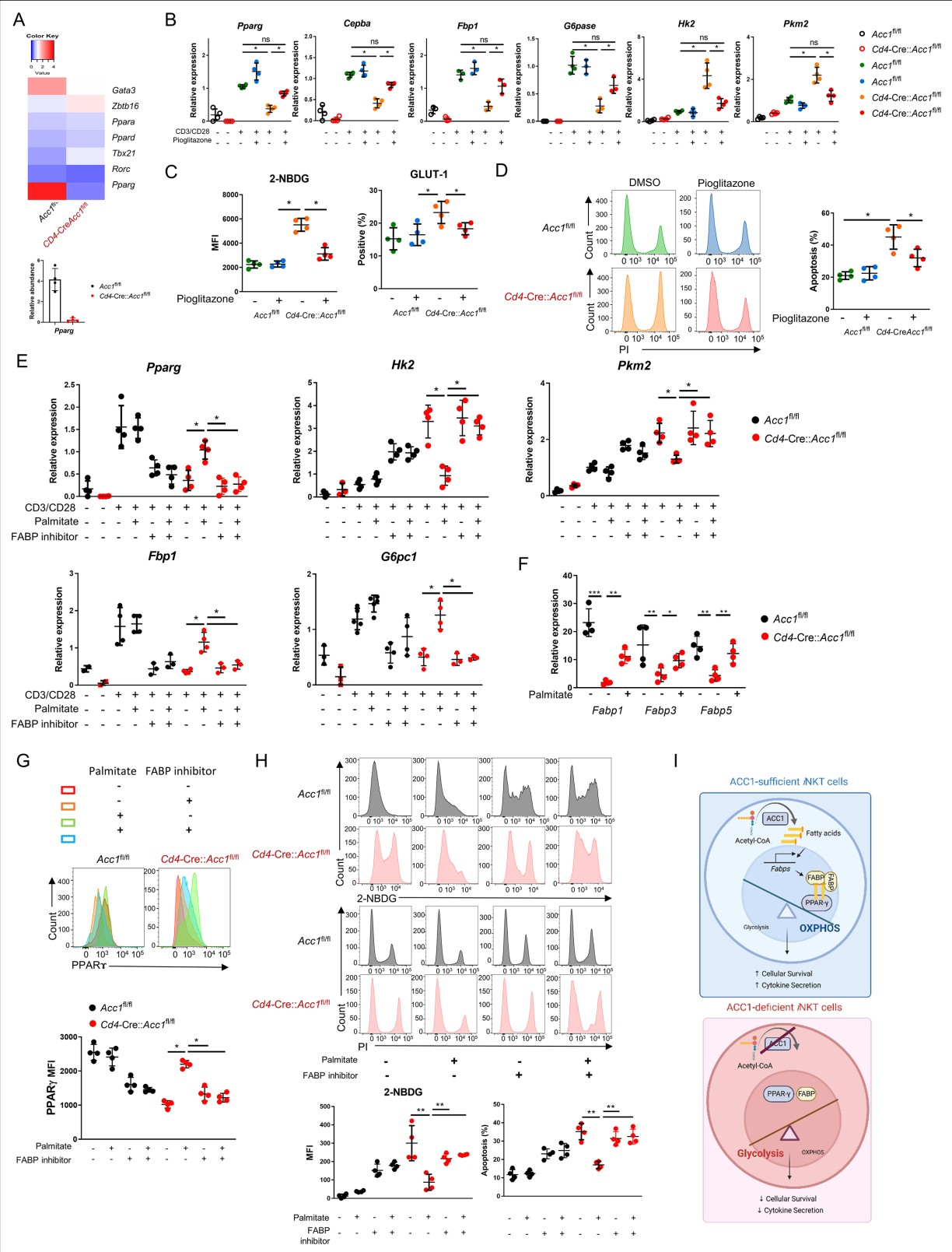

**Figure 5.** Low PPARγ expression in ACC1-deficient invariant natural-killer T (*i*NKT) cells contributes to their enhanced cell death. *i*NKT cells sorted from the liver of naïve *Cd4*-Cre::*Acc1*^fl/fl and *Acc1*^fl/fl mice were either used directly in the experiment (**A**) or cultured with and/or without anti-CD3/CD28 with and without the indicated additives (**C–H**). (**A**) Heat map of the transcription factor genes and RT-PCR confirmation of *Pparg* expression in the uncultured *i*NKT-cell preparations. (**B**) CD3/CD28-stimulated and unstimulated *i*NKT cells were cultured with and without pioglitazone and *Pparg*, *Cepba*, *Fbp1*,

*Figure 5 continued on next page*

*Figure 5 continued*

*G6pase*, *Hk2*, and *Pkm2* expression was examined. (**C, D**) Unstimulated *i*NKT cells were cultured with and without pioglitazone and 2-NBDG uptake, GLUT1 expression (**C**), and cell death (**D**) were examined. (**E**) CD3/CD28-stimulated and unstimulated *i*NKT cells were cultured with and without palmitate and/or FABP inhibitor and *Pparg*, *Cepba*, *Fbp1*, *G6pase*, *Hk2*, and *Pkm2* expression was examined. (**F**) Unstimulated *i*NKT cells were cultured with and without palmitate and *Fabp1*, *Fabp3*, and *Fabp5* expression was examined. (**G, H**) Unstimulated *i*NKT cells were cultured with and without palmitate and/or FABP inhibitor and PPARγ expression (**G**), 2-NBDG uptake, and cell death (**H**) were examined. (**I**) Graphical abstract of ACC1-FABP-PPARγ in *i*NKT cells are shown. The data in (**B–H**) are pooled data from four independent experiments (n = 4/group) and are presented as mean ± SEM. ns, not significant. *p<0.05, **p<0.01, ***p<0.001, as determined by unpaired two-tailed *t*-tests.

The online version of this article includes the following figure supplement(s) for figure 5:

**Figure supplement 1.** ACC1 deficiency does not affect epigenetic modification of *Pparg* in invariant natural-killer T (*i*NKT) cells.

Since FABP-inhibitor treatment did not affect the PPARγ expression, glucose uptake, and cell death in WT *i*NKT cells (*Figure 5E, G, and H*), it seems that the ACC1 deficiency in *i*NKT cells mediated their metabolic reprogramming by abolishing palmitate production: this downregulated their FABP expression, which in turn suppressed PPARγ expression, which increased glycolysis and thereby promoted cell death (*Figure 5I*).

On a side note, the WT and ACC1-deficient *i*NKT cells did not differ in terms of the methylation of the promoter site of PPARγ (*Figure 5—figure supplement 1A*), and treatment of the unstimulated cells with histone deacetylase (HDAC) inhibitor did not alter their cell-death rates or expression of *Pparg*, *Hk2*, *Bcl2*, or *Bak* (*Figure 5—figure supplement 1B, C*). Thus, the regulation of PPARγ expression in ACC1-deficient *i*NKT cells may not involve epigenetic changes such as methylation and acetylation.

## The ACC1-PPAR-γ axis in *i*NKT cells contributes to AHR and airway inflammation in allergic asthma

To confirm that ACC1-mediated metabolic programming in *i*NKT cells regulates the development of AHR, we pretreated WT or ACC1-deficient *i*NKT cells with palmitate or pioglitazone, adoptively transferred them into Jα18 KO mice, and then subjected the mice to OVA-induced asthma (*Figure 6*) or HDM-induced asthma (Fig. S4). As shown above (*Figure 3*), transfer of untreated WT *i*NKT cells was able restore AHR, airway inflammation scores, and high lymphocyte, neutrophil, eosinophil, and macrophage frequencies in the lungs and BALF. It also generated a CD4+ T cell population in the lungs that expressed high levels of *Gata3*, *Il4*, *Il5*, and *Il13* and low levels of *Tbx21*, *Rorc*, *Foxp3*, *Ifng*, *Il17a*, and *Il10*. Moreover, the sorted *i*NKT cells in the lungs expressed high levels of *Il4* and *Il13* but low levels of *Ifng* and *Il17a*. By contrast, transfer of the untreated ACC1-deficient *i*NKT cells did not achieve any of these pathological changes or lung CD4+ T cell or *i*NKT cell gene-expression profiles. However, when the ACC1-deficient *i*NKT cells were pretreated with palmitate or pioglitazone, all pathological and gene-expression changes were restored (*Figure 6A–G* and *Figure 6—figure supplement 1*).

As shown above (*Figure 3I*), the transfer of untreated WT *i*NKT cells into mice that were then subjected to OVA-induced asthma led to a moderate *i*NKT cell population in the lungs that showed low apoptosis. By contrast, the transfer of untreated ACC1-deficient *i*NKT cells led to low lung *i*NKT cell numbers that frequently took up PI. However, pretreatment of the ACC1-deficient *i*NKT cells with palmitate or pioglitazone increased lung *i*NKT cell numbers and reduced their apoptosis (*Figure 6H and I*). This is consistent with our in vitro experiments with the unstimulated and anti-CD3/CD28-stimulated *i*NKT cells above (*Figure 5*). Notably, the sorted *i*NKT cell population in the lungs of Jα18 KO mice that received untreated ACC1-deficient *i*NKT cells also demonstrated less PPARγ expression and more glucose uptake and Glut1 expression compared to when untreated WT *i*NKT cells were transferred, and this was reversed when the ACC1-deficient *i*NKT cells were pretreated with palmitate or pioglitazone (*Figure 6J*). Thus, treatment with palmitate or pioglitazone increased the survival of adoptively transferred ACC1-deficient *i*NKT cells in Jα18 KO mice, which restored their ability to induce AHR in allergic asthma. Meanwhile, chemokine receptor signaling is also implicated in regulating homeostasis of *i*NKT cell in the periphery. In particular, Meyer et al. suggested that *i*NKT cells require CCR4 to localize to the airways and to induce AHR (*Meyer et al., 2007*). Thus, we examined the expression of several chemokine receptors, including CCR4. We found that WT and ACC1-deficient *i*NKT cells did not differ in their chemokine receptor expressions, suggesting that the chemokine signaling may not be critical for ACC1-mediated regulation in AHR.

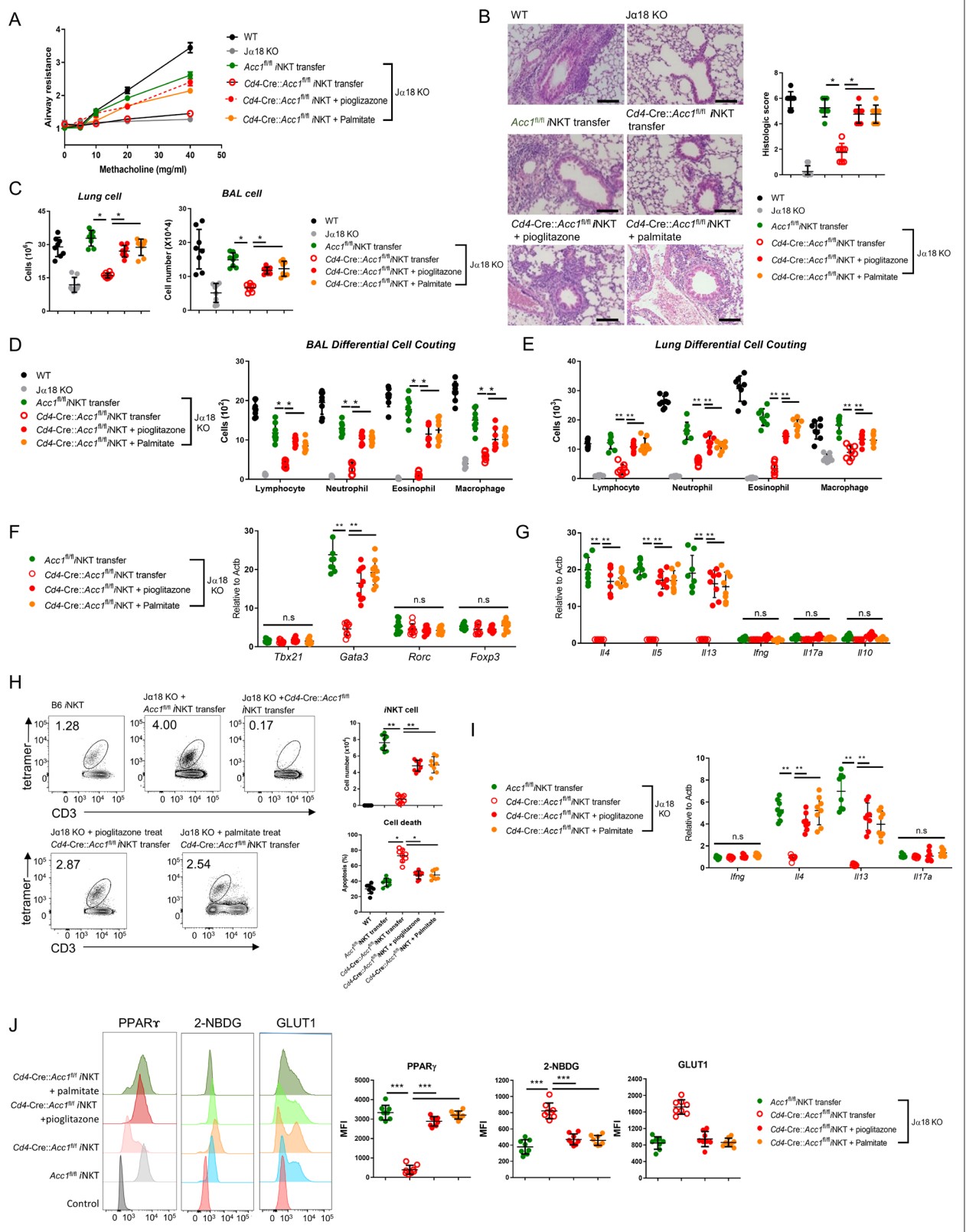

**Figure 6.** Pretreatment of ACC1-deficient invariant natural-killer T (iNKT) cells with PPARγ agonist or exogenous fatty acid restores ovalbumin (OVA)-induced asthma. iNKT cells sorted from naïve *Cd4*-Cre::*Acc1*fl/fl and *Acc1*fl/fl mice were pretreated with and without pioglitazone or palmitate and then adoptively transferred into naïve WT and Jα18 KO mice. OVA-induced asthma was then initiated. (**A**) Airway resistance. (**B**) Representative histological images and histological scores of lung sections. Scale bar = 200 μM. (**C**) Total immune cell numbers in the lungs and bronchoalveolar lavage fluids

*Figure 6 continued on next page*

*Figure 6 continued*

(BALFs). (**D, E**) Immune-cell subsets in the BALFs (**D**) and lungs (**E**). (**F, G**) Transcription factor (**F**) and cytokine (**G**) expression in the total CD4$^+$ T cell population from the lungs. (**H**) Cell numbers and survival status of the lung *i*NKT cells. (**I**) Cytokine expression in the lung *i*NKT cells. (**J**) PPARγ, 2-NBDG, and GLUT1 expression in the lung *i*NKT cells. The data are pooled from four independent experiments (n = 8/group) and are presented as mean ± SEM. ns, not significant. *p<0.05, **p<0.01, ***p<0.001, as determined by unpaired two-tailed *t*-tests.

The online version of this article includes the following figure supplement(s) for figure 6:

**Figure supplement 1.** Adoptive transfer of Acc1-deficient invariant natural-killer T (*i*NKT) cells pretreated with exogenous fatty acid or PPAR-γ agonist into *i*NKT cell-deficient mice (Jα18 KO) restores AHR in the house-dust mite (HDM)-induced asthma model.

**Figure supplement 2.** Genes for fatty acid metabolism are highly expressed in invariant natural-killer T (*i*NKT) cells from peripheral blood mononuclear cells (PBMCs) of patients with allergic asthma.

Altogether, these results show that ACC1-mediated metabolic programming in *i*NKT cells maintains their PPARγ expression, which downregulates their glycolysis and thereby increases their survival. This allows these cells to promote AHR and airway inflammation in allergic asthma models.

## The ACC1-PPARγ axis in *i*NKT cells may also contribute to allergic asthma in humans

We next asked whether patients with allergic asthma demonstrate altered expression of fatty-acid synthesis-related metabolic enzymes in *i*NKT cells compared to healthy volunteers. Thus, we sorted the *i*NKT, Treg, CD4 +T, and CD8$^+$ T cells from the peripheral blood mononuclear cells (PBMCs)of 10 healthy volunteers, 10 patients with allergic asthma, and 10 patients with nonallergic asthma and examined their mRNA expression of *ACC1* and *PPARG*. We also examined the expression of *FASN* because it, like ACC1, is a key de novo fatty-acid synthesis enzyme and our experiment with CD3/CD28-stimulated murine *i*NKT cells and conventional CD4$^+$ and CD8$^+$ T cells showed that the *i*NKT cells expressed much higher levels of *Fasn* as well as *Acc1* than the other cells (***Figure 1C***). Indeed, circulating *i*NKT cells from allergic asthma patients expressed significantly higher levels of *ACC1*, *FASN*, and *PPARG* than *i*NKT cells from healthy controls and nonallergic asthma patients (***Figure 6—figure supplement 2A***). Similarly, the *i*NKT cells from the allergic asthma patients expressed significantly higher *FBP1* levels than the control *i*NKT cells, although this difference was not observed for *HK2* (***Figure 6—figure supplement 2B***). Lastly, the expression levels of *IL4* and *IL13* were significantly higher in *i*NKT cells from the allergic asthma patients compared to those from healthy controls and nonallergic asthma patients (***Figure 6—figure supplement 2C***). These changes in *ACC1*, *FASN*, *PPARG*, and *FBP1* expression were much more pronounced in the *i*NKT cells from allergic asthma patients than in the CD4$^+$ T cells, CD8$^+$ T cells, and Tregs from the same patients (***Figure 6—figure supplement 2***). Thus, *i*NKT cells from allergic asthma patients express higher *ACC1*, *FASN* and *PPARG* levels and lower levels of a glycolysis, which is accompanied with higher levels of *IL4* and *IL13* than *i*NKT cells from healthy controls and nonallergic asthma patients. These findings suggest that the ACC1-PPAR-γ axis in *i*NKT cells may also be involved in human allergic asthma.

## Discussion

This study showed that *Cd4*-Cre::*Acc1*$^{fl/fl}$ mice exhibited attenuated AHR and airway inflammation in OVA- and HDM-induced asthma models due to the enhanced apoptosis and loss of *Gata3*, *Il4*, and *Il13* expression in lung *i*NKT cells. The mechanism appeared to partly involve the low production in *i*NKT cells of the ACC1-downstream long-chain fatty acid palmitate, which impaired FABP expression; this reduced PPARγ expression and enhanced glycolysis in the *i*NKT cells, which promoted their apoptosis. Moreover, the *Cd4*-Cre*Acc1*$^{fl/fl}$ mice displayed significant defects in the thymic development and peripheral homeostasis of *i*NKT cells in a cell-intrinsic manner; thus, homeostatic dysregulation of *i*NKT cells may also contribute to the attenuation of AHR in *Cd4*-Cre::*Acc1*$^{fl/fl}$ mice.

The negative effect of ACC1 deficiency on the asthmogenic *i*NKT-cell functions is consistent with several studies that suggest that ACC1-mediated de novo fatty-acid synthesis can also shape the behavior of other T cells, including Th17 cell and Treg differentiation, IL-5 production by Th2 cells in the lungs, the generation of memory CD4$^+$ T cells, and CD8$^+$ T cell proliferation. Like us, these studies also show that blocking ACC1 function can improve various diseases in mouse models (***Berod et al., 2014***; ***Nakajima et al., 2021***; ***Endo et al., 2019***; ***Goto et al., 2014***). In this regard, Tregs may also

play a major role in asthma. However, the expression level of Foxp3 was comparable between WT and ACC1-deficient Tregs. The level of Foxp3 to some extent serves as a critical determinant of suppressive function of Tregs (*Fontenot et al., 2003*; *Wing et al., 2019*). Thus, we speculate that they might not critically contribute to the development of asthma, although we cannot completely rule out the contribution of Tregs to our studies.

It should be noted that *Cd4*-Cre::*Acc1*^fl/fl mice lack ACC1 expression in both conventional CD4$^+$ T cells and *i*NKT cells. While the use of *i*NKT cell-specific Cre system would demonstrate critical role of ACC1 in *i*NKT cells regarding allergic asthma, there is no *i*NKT cell-specific Cre system available yet. In addition, the study conducted by Nakajima et al., which reported that the absence of ACC1 in CD4$^+$ T cells resulted in reduced numbers and functional impairment of memory CD4$^+$ T cells, leading to less airway inflammation further suggests the possibility of involvement of conventional CD4$^+$ T cells in regulation of allergic asthma.

However, based on our experimental results, we believe that *i*NKT cells more contribute to the regulation of allergic asthma for the following reasons: (i) while the number of *i*NKT cells was significantly reduced in *Cd4*-Cre::*Acc1*^fl/fl mice, the number of conventional CD4$^+$ T cells was only slightly reduced, (ii) *Cd4*-Cre::*Acc1*^fl/fl mice were dramatically decreased in their AHR in α-GalCer-induced allergic asthma model, and (iii) Jα18 KO mice that lack *i*NKT cells almost completely restore their AHR when adoptively transferred with WT *i*NKT cells but not ACC1-deficient *i*NKT cells. These results indicate that ACC1-mediated regulation of AHR is significantly dependent on *i*NKT cells, which might contribute to AHR in the study conducted by Nakajima et al. as well. From these, we believe that while ACC1 is a critical regulator of both conventional CD4$^+$ T cells and *i*NKT cells in the regulation of allergic asthma, *i*NKT cells may contribute more to the regulation of allergic asthma compared to CD4$^+$ T cells.

The possibility that ACC1 regulates survival of *i*NKT cells by programming their metabolism was first revealed by our transcriptome analysis of treatment-naïve *i*NKT cells: the ACC1-deficient *i*NKT cells exhibited downregulation of fatty-acid synthesis genes, as could be expected, but also upregulation of glycolysis pathway-related genes. This impaired the survival of the cells: compared to the WT *i*NKT cells, the ACC1-deficient *i*NKT cells were prone to apoptosis. This was observed in ACC1-deficient *i*NKT cells regardless of whether they were treatment-naïve, stimulated by TCR ligation in vitro, or activated during OVA/HDM-mediated asthma. And the apoptotic tendency of these cells was completely reversed by treatment with glycolysis inhibitor in vitro. These findings are consistent with those of. *Kumar et al., 2019*, who reported that the production of lactate by *i*NKT cells when their glycolysis is upregulated promotes their cell death.

Furthermore, the apoptotic tendency of the ACC1-deficient *i*NKT cells was accompanied by their functional impairment. The ACC1-deficient *i*NKT cells exhibited impaired viability and functionality. Treatment of glycolysis inhibitor in ACC1-deficient *i*NKT cells not only restored cellular survival but also their functionalities. From these results, we speculate that ACC1-mediated regulation of both cellular homeostasis and cytokine production cooperatively contributed to the asthma phenotype.

Our transcriptome assays of naïve *i*NKT cells showed that of the multiple candidate transcription factors that were examined, only *Pparg* was upregulated in WT *i*NKT cells but not ACC1-deficient *i*NKT cells. This difference was augmented by TCR ligation. This suggested that downregulation of PPARγ could mediate the enhanced glycolysis and apoptosis in ACC1-deficient *i*NKT cells. This was supported by the fact that the PPARγ agonist pioglitazone largely restored gluconeogenesis and downregulated glycolysis and apoptosis in these cells. These findings are consistent with reports that PPARγ regulates the glycolytic pathway, although heterogeneous effects have been noted (*Lehmann et al., 1995*; *Masters et al., 1987*; *Roberts et al., 2011*; *Panasyuk et al., 2012*; *Coman et al., 2016*; *Calvier et al., 2017*). Our study also supports Fu et al., who observed that lipid biosynthesis is increased in *i*NKT cells after activation, and that this is mediated by PPARγ. Specifically, PPARγ acts synergistically with the transcription factor PLZF (which drives the acquisition of T-helper effector functions by innate and innate-like lymphocytes) to activate the *Srebf1* transcription factor (*Mao et al., 2016*), which regulates lipid biosynthesis. This induces cholesterol synthesis, which is required for the optimal production of IFNγ by the *i*NKT cells. This PPARγ/PLZ-*Srebf1*-cholesterol pathway was found to contribute to anti-tumor immunity (*Fu et al., 2020*). Previous studies also reported that the PPARγ agonist ciglitazone promotes the TGFβ-dependent conversion of naïve effector T cells into Tregs, whereas PPARγ deficiency increases the infiltration of Th17 cells into the central nervous system in

experimental autoimmune encephalomyelitis. (*Wohlfert et al., 2007*; *Klotz et al., 2009*). Thus, PPARγ can regulate the differentiation of naïve CD4⁺ T cells into both Th17 or Treg cells, which supports the role of PPARγ in *i*NKT-cell homeostasis and functions. However, further research is needed to determine the mechanisms by which PPARγ regulates the metabolic programming and biological changes in *i*NKT cells.

We also explored how PPARγ expression was downregulated in ACC1-deficient *i*NKT cells. Several studies have reported that PPARγ expression can be regulated by endogenous ligands such as FABPs, as well as by histone methylation and acetylation (*Huang et al., 2018*; *Furuhashi and Hotamisligil, 2008*), and that these changes can be regulated by the intracellular lipid levels (*Veerkamp and van Moerkerk, 1993*; *Barak and Lee, 2010*). Indeed, we observed that (i) ACC1-deficient *i*NKT cells demonstrated low FABP expression, (ii) exogenous fatty acid not only reversed the effects of ACC1 deficiency on glycolysis and apoptosis, it also restored the expression of both FABPs and PPARγ, and (iii) treating ACC1-deficient *i*NKT cells with exogenous palmitate or a PPARγ agonist and then transferring them into *i*NKT cell-deficient mice restored AHR in OVA- or HDM-induced asthma models. Thus, the reduced de novo fatty-acid synthesis in ACC1-deficient *i*NKT cells downregulated the expression of FABPs, which decreased the expression and activity of PPARγ, which in turn promoted glycolysis and apoptosis. This FABP-PPARγ axis was also observed in studies on hepatocytes and adipocytes, which reported that FABPs interact with PPARγ, thereby inducing its transactivation (*Wolfrum et al., 2001*; *Tan et al., 2002*). However, we did not observe that PPARγ promoter methylation was altered in the ACC1-deficient *i*NKT cells. Taken together, our findings indicate that the ACC1-FABP-PPARγ axis drives the proglycolytic metabolic reprogramming in ACC1-deficient *i*NKT cells, thereby downregulating *i*NKT-cell homeostasis, and functions, including *i*NKT cell-mediated AHR in asthma models.

Finally, we showed that the ACC1-FABP-PPARγ axis in *i*NKT cells may also play pathogenic roles in human allergic asthma: *i*NKT cells in the blood of patients with allergic asthma expressed higher levels of *IL4*, *IL13*, *ACC1*, *FASN*, and *PPARG* and lower levels of a glycolytic gene than the *i*NKT cells of nonallergic asthma patients and healthy controls. Moreover, these genes were up/downregulated much more strongly in the *i*NKT cells from allergic asthma patients than in the other T-cell subsets from the same patients. Notably, the differences between allergic asthma patients and the control patients did not reflect lower frequencies of *i*NKT cells in the PBMCs from the allergic asthma patients (data not shown). This is consistent with speculation in the literature that *i*NKT cells participate in human asthma by altering their phenotype rather than their frequencies: for example, although asthma patients and controls do not differ in terms of *i*NKT-cell frequencies in the BALF, the asthma *i*NKT cells express more IL-4 (*Akbari et al., 2006*; *Shim and Koh, 2014*). Thus, the ACC1-FABP-PPARγ axis in *i*NKT cells may be activated and thereby contribute to human allergic asthma pathology.

In conclusion, derangement of the ACC1-FABP-PPAR-γ axis induces glycolysis-related impairment of cell viability in *i*NKT cells, which limits their ability to induce AHR and airway inflammation in murine asthma models. Therefore, de novo lipid synthesis in *i*NKT cells may play a critical regulatory role in the development of AHR.

## Materials and methods

### Mice

Female C57BL/6 (B6) mice were obtained from the Orient Company (Seoul, Korea). *Cd4*-Cre mice were purchased from the Jackson Laboratory (stock no. 017336; The Jackson Laboratory, Bar Harbor, ME). *Acc1*^fl/fl^ mice were originally generated and kindly provided by Dr. Salih J. Wakil of the Baylor College of Medicine (Houston, TX). To generate *Cd4*-Cre::*Acc1*^fl/fl^ mice, the *Acc1*^fl/fl^ mice were crossed with *Cd4*-Cre mice. The animals were bred and maintained under specific-pathogen-free conditions at the Biomedical Research Institute of Seoul National University Hospital (Seoul, Korea). All experiments were conducted on mice that were 8–10 weeks old. All animal experiments were approved by the Institutional Animal Care and Use Committee at Seoul National University Hospital (SNUH-IACUC). The animals were maintained in an AAALAC International (#001169)-accredited facility in accordance with the Guide for the Care and Use of Laboratory Animals 8th edition.

## Isolation of PBMCs from healthy subjects and asthmatic patients

In total, 30 human PBMC samples were obtained from healthy control subjects and patients with nonallergic and allergic asthma. The PBMCs were obtained from 30 mL peripheral blood by using Ficoll-Paque (GE Healthcare, Uppsala, Sweden). The $i$NKT, CD4$^+$ T, Foxp3$^+$ Tregs, and CD8$^+$ T cells in the PBMCs were sorted by FACSAria II (Becton Dickinson, San Jose, CA; purity <95%). All subjects provided informed written consent to participate in the study. The Institutional Review Board of Seoul National University Bundang Hospital approved the human studies (IRB #: B-1901/517-304).

## Preparation and activation of murine T and $i$NKT cells

To determine the mRNA expression, transcriptome, metabolome, ECAR, and OCR of T and $i$NKT cells, the cells were obtained from mice as follows. CD4$^+$ or CD8$^+$ T cells from the thymi, spleens, lungs, and livers of $Acc1^{fl/fl}$ and $Cd4$-Cre::$Acc1^{fl/fl}$ mice were enriched by FACSAria II (Becton Dickinson). To obtain $i$NKT cells, the mononuclear cells of the livers of $Acc1^{fl/fl}$ and $Cd4$-Cre$Acc1^{fl/fl}$ mice were isolated by Percoll gradient centrifugation. Thereafter, the $i$NKT cells were isolated by using APC-conjugated α-GalCer/CD1d tetramers-unloaded or loaded with PBS-57 from the tetramer facility of the National Institutes of Health (Bethesda, MD) followed by sorting using FACS (purity <95%). Alternatively, thymic and liver mononuclear cells were labeled with APC-conjugated α-GalCer/CD1d tetramers, bound to anti-APC magnetic beads, and enriched on a MACS separator (Miltenyi Biotec, Auburn, CA; purity 89%). To analyze the development of thymic $i$NKTs cells, we re-stained enriched cells with CD1d tetramer and gated out CD3 and CD1d tetramer double-positive cells via flow cytometry to identify thymic $i$NKT cells, which were used for further analysis. For functional analysis, $i$NKT, CD4$^+$ T, and CD8$^+$ T cells were stimulated with plate coated anti-CD3 and CD28 antibodies overnight.

## Adoptive transfer of $i$NKT cells in allergic asthma models

$i$NKT cells were obtained from the lungs of at least 10 $Acc1^{fl/fl}$ or $Cd4$-Cre::$Acc1^{fl/fl}$ mice. Mouse lungs were finely chopped into small pieces using razor blades and were enzymatically digested using type IV collagenase. $i$NKT cells from the lungs were sorted via FACS using CD1d tetramers. Approximately, $6.0 \times 10^5$ of $i$NKT cells were obtained from at least 10 mice and were adoptively transferred into individual recipient mouse via the intratracheal route.

## Antibodies, reagents, and preparation of fatty acids

The antibodies used in flow cytometry are summarized in *Supplementary file 1*. The PPARɣ agonist pioglitazone (E6910), the histone deacetylase (HDAC) inhibitor MG-132 (474790), 2-deoxy-D-glucose (2-DG, D6134), and palmitic acid (C16:0, P9767) were purchased from Sigma-Aldrich (St. Louis, MO). Briefly, palmitic acid was dissolved in sterile water using a vortex and heated to 70°C for 10 min. Palmitic acid was conjugated to bovine serum albumin (BSA, Biosesang, Seongnam, Korea) in serum-free RPMI containing 5% non-esterified fatty acids (NEFA)-free BSA immediately after dissolving as described previously (*Mayer and Belsham, 2010*). The conjugated-palmitic acid was shaken at 140 rpm at 40°C for 1 hr before being added to the cells. Serum-free RPMI containing 5% NEFA-free BSA was used as the vehicle control. Cells were treated with pioglitazone at 10 μM, 2-DG at 10 mM, MG-132 at 5 μM, or palmitate at 100 μM.

## Flow-cytometric analysis

Single-cell suspensions were preincubated with mouse anti-Fc receptor antibodies (BD Biosciences, Franklin Lakes, NJ), stained with antibodies for 30 min at 4°C, washed twice with 1× PBS, and analyzed using a BD LSRII flow cytometer (BD Biosciences). For intracellular cytokine staining, cells were stained for surface markers, permeabilized, and then stained for intracellular proteins. Cells were permeabilized and fixed with Intracellular Fixation & Permeabilization Buffer (eBioscience, Waltham, MA), according to the manufacturer's protocols. Data were analyzed with FlowJo software (version 10; TreeStar, Ashland, OR).

## Measurement of glucose uptake capacity using 2-NBDG assay

To assess the uptake of the fluorescent glucose analog 2-NBDG, cells were incubated for 30 min with 50 μM 2-NBDG (Cambridge Bioscience, Cambridge, UK)-supplemented glucose-free RPMI 1640 or RPMI containing 10% (v/v) FBS supplemented with 12.5 nM MitoTracker Deep Red (Thermo Fisher

Scientific, Waltham, MA) for 30 min at 37°C, 5% (v/v) $CO_2$. The cells were washed with 1× PBS, fixed with 0.5% (w/w) paraformaldehyde (PFA, Sigma), and then captured on a BD LSR Fortessa flow cytometer (BD Biosciences). After treating 2-NBDG, the fluorescence intensity of cells were measured using flow cytometry and represented the degree of glucose uptake in cells.

### Generation of mixed BM chimeras

BM cells were prepared from the femurs and tibias of WT B6 (CD45.1[+] background) or *Cd4*-Cre::*Acc1*[fl/fl] B6 (CD45.2[+] background) donor mice. Recipient mice (CD45.2[+] background) were lethally irradiated (800 rads) and injected intravenously with a 1:1 mixture of BM cells from WT and *Cd4*-Cre::*Acc1*[fl/fl] mice (1 × 10[6] cells) in total. The chimeras were analyzed 8 wk after BM transplantation.

### OVA- and HDM-induced asthma models

To induce the OVA asthma model, OVA antigen (Sigma) was dissolved in saline to a 1 mg/mL concentration. The mice were injected intraperitoneally with 4 mg aluminum hydroxide (Thermo Fisher Scientific) mixed with 100 μL OVA in saline (1 mg/mL solution) on days 0 and 7. On days 14, 15, 16, and 17, the isoflurane-anesthetized mice were challenged with 50 μg OVA (1 mg/mL solution) intranasally. Twenty-four hours after the last OVA instillation, the mice were sacrificed to measure the AHR. To induce the HDM asthma model, HDM (B82; Greer) was dissolved in saline to a final concentration of 1 mg/mL. Isoflurane-anesthetized mice were then treated intranasally on days 1–14 with 25 μg HDM solution and, 96 hr after the final intranasal instillation, the AHR and other inflammatory variables were measured.

### In vivo intranasal inoculation of α-GalCer

Mice were inoculated intranasally with 200 ng of α-GalCer dissolved in 50 μL PBS and, 24 h after the last instillation, the mice were sacrificed to measure the AHR.

### Measurement of AHR

AHR was measured by administering increasing doses of methacholine (0, 5, 10, 20, and 40 mg/mL) using the forced oscillation technique (FlexiVent System; SCIREQ, Montreal, Quebec, Canada). A snapshot perturbation maneuver was imposed to measure the resistance of the entire respiratory system.

### Preparation of single-cell suspensions from lungs and BALF and differential counting

Mouse lungs were diced into small pieces using razor blades and incubated for 1 hr in digestion medium (RPMI-1640 with 1 mg/mL collagenase type 4, Sigma) at 37°C. The lung cells were then filtered and treated with red blood cell lysis buffer. BALF cells were acquired by administering 1 mL PBS supplemented with 2% FBS three times using a tracheal cannula. Lavage fractions were pooled together for total and differential cell counts. Differential cell counts were conducted with cytospin preparations after Giemsa staining and were validated by flow cytometric analysis using a BD Fortessa (BD Biosciences).

### Transcriptome analysis

*i*NKT cells from *Acc1*[fl/fl] and *Cd4*-Cre::*Acc1*[fl/fl] mice were sorted and total RNA was extracted using RNeasy with QIAshredders (QIAGEN, Germantown, MD). RNA quality and quantity were determined using an Agilent 2100 Bioanalyzer (Agilent Technologies, Santa Clara, CA) and Nanodrop (Nanodrop Technologies, Wilmington, DE). RNA-seq analyses were performed at the Theragen Bio Institute (Suwon, Korea). The data are available in NCBI database with accession number GSE205761. GSEA gene set enrichment assay (GSEA) was performed to identify significant expression differences under the two biological conditions and to identify the gene classes that were overexpressed or underexpressed in *i*NKT cells from *Cd4*-Cre::*Acc1*[fl/fl] mice relative to those from WT mice. The transcriptome data were from sorted *i*NKT cells from eight mice/group that were pooled together. We also analyzed transcriptome data from our previous study to compare the *i*NKT cells to conventional CD4[+]T cells (GSE103190) (*Oh et al., 2011*). These analyses were performed using GenePattern (https://genepattern.broadinstitute.org/).

## Metabolome analysis

*i*NKT cells were sorted and pooled from eight WT mice, or eight *Cd4*-Cre::*Accl1*$^{fl/fl}$ mice. The metabolites were extracted from each cell group as previously described (*Faubert et al., 2013*; *Faubert et al., 2014*), followed by MeOX and MTBSTFA derivatization. Metabolic profiling was performed by GC-MS/MS dynamic multiple reaction monitoring (dMRM) analyses using the Agilent 7890/7000 GC triple quadrupole mass spectrometer system. Gas chromatography was conducted using an Agilent J&W HP-5ms UI 15 m × 0.25 mm × 0.25 µm (P/N 19091S-431UI) capillary column. Helium was used as the carrier gas with a constant column flow rate of 1.5 mL min$^{-1}$. The initial oven temperature of 60°C was increased to 320°C at a rate of 10°C min$^{-1}$. The mass spectrometer was operated in the electron ionization mode at 70 eV. The analyses were performed in the dMRM mode. The area ratios of the analyte and the internal standard were used for the evaluation. The split/splitless injector temperature was set to 280°C. The samples were injected in the splitless mode with an autosampler Agilent 7683A injector. Agilent MassHunter Data Acquisition Software (ver. B.04.00) and MassHunter Workstation Software for Quantitative Analysis (QQQ) were used for data acquisition and the quantitative analysis, respectively. The results were normalized to the *i*NKT-cell numbers and/or total protein concentrations.

## Quantitative RT-PCR

To analyze gene-expression levels in the metabolic pathways, immune cells were sorted and pooled from four-eight mice/group, and used in the analysis. Total RNA was isolated with TRIzol reagent (Thermo Fisher Scientific) according to the manufacturer's protocol. RNA was reverse-transcribed into cDNA using Maloney murine leukemia virus reverse transcriptase Taq polymerase (Promega, Madison, WI). For quantitative RT-PCR, gene-specific PCR products were quantified by an Applied Biosystems 7500 Sequence Detection System (Applied Biosystems, Foster City, CA). The list of primers used and their sequences is shown in *Supplementary file 2*. Gene-expression levels were normalized to those of *Actb* for mice and *ACTB* for humans.

## Histology

Lung tissues were obtained from WT and Jα18 KO mice that did or did not undergo transfer of *i*NKT cells from *Acc1*$^{fl/fl}$ and *Cd4*-Cre::*Acc1*$^{fl/fl}$ mice during OVA- or HDM-induced asthma. The tissues were fixed in 10% formalin, embedded in paraffin, and sectioned, followed by staining with hematoxylin and eosin or periodic acid-Schiff (PAS). Lung inflammation and goblet cell hyperplasia were graded using a semi-quantitative scoring system as previously described( *Kujur et al., 2015*; *Myou et al., 2003*). Briefly, inflammatory-cell numbers were graded as follows: 0, normal; 1, few cells; 2, a ring of inflammatory cells that was 1 cell-layer deep; 3, a ring of inflammatory cells that was 2–4 cell-layers deep; and 4, a ring of inflammatory cells that was >4 cell-layers deep. The goblet-cell numbers in the airway were graded as follows: 0, <0.5% PAS-positive cells; 1, <25%; 2, 25–50%; 3, 50–75%; and 4, >75%. Two fields were counted for each slide and mean score was calculated for each group. Two pulmonary pathologists reviewed and analyzed representative images.

## ECAR and OCR assays

*i*NKT cells were sorted from eight mice/group and pooled together. ECAR and OCR assays were performed using a Seahorse Extracellular Flux Analyzer XF24e (Seahorse Bioscience Inc, Santa Clara, CA) according to the manufacturer's instructions. Briefly, the cells were seeded in a Seahorse plate and cultured overnight to 70–80% confluence. The culture medium was then replaced with cellular assay medium supplemented with 1 mM glutamine for the ECAR assay and 1 mM pyruvate, 2 mM glutamine, and 10 mM glucose for the OCR assay. The cells were incubated for 1 hr in a $CO_2$-free incubator before measurement. The assays were performed according to the Seahorse protocols with final concentrations of 10 mM glucose, 2 µM oligomycin, and 50 mM 2-DG for ECAR, and 2 µM oligomycin, 1 µM carbonyl cyanide-4 (trifluoromethoxy) phenylhydrazone (FCCP), and 1 µM antimycin A for OCR. These assays were performed in at least three independent experiments on separate days.

## Pyrosequencing assay

Genomic DNA was converted with bisulfite by using the EZ DNA Methylation Kit (Zymo Research, Irvine, CA) according to the manufacturer's protocol. PyroMark software was used to design the pyrosequencing primers. In brief, bisulfite-converted DNA was subjected to PCR amplification in

a total volume of 25 µL. DNA methylation was assessed using a PyroMark Q48 Autoprep system (QIAGEN) and PyroMark Q48 Advanced CpG Reagents (QIAGEN). The nucleotide dispensation order was generated by entering the sequences into the PyroMark Q48 Autoprep software ver. 2.4.2 (QIAGEN). The methylation at each CpG site was determined using the PyroMark Q48 Autoprep software set in CpG mode. The mean methylation of all CpG sites within the target region was determined by using the methylation at the individual CpG sites.

## Materials availability statement

RNA-sequencing of *i*NKT cells from *Acc1*<sup>fl/fl</sup> and *Cd4*-Cre::*Acc1*<sup>fl/fl</sup> mice are available in NCBI database with accession number GSE205761. Transcriptome data from our previous study comparing *i*NKT cells to conventional CD4$^+$T cells are available in NCBI dataset with accession number GSE103190.

## Statistical analyses

GraphPad Prism 7 (GraphPad Software, La Jolla, CA) was used for the statistical analyses and graphical display of the data. Unpaired two-tailed *t*-tests or Mann–Whitney *U* test were performed to compare the groups. p-values <0.05 were considered statistically significant.

## Acknowledgements

We would like to appreciate the NIH Tetramer Core Facility at the National Institute of Health, USA, for providing PBS-57 loaded CD1d tetramers. This work was supported by the Basic Research Program through the National Research Foundation of Korea (NRF) funded by the MSIT (NRF-2020R1A4A1017515) and the National Research Foundation of Korea (NRF) grant funded by the Korea government (MSIT) (NRF-2020R1A2C2008312).

## Additional information

### Funding

| Funder | Grant reference number | Author |
| --- | --- | --- |
| National Research Foundation of Korea | NRF-2020R1A4A1017515 | Doo Hyun Chung Jaemoon Koh Yeon Duk Woo |
| National Research Foundation of Korea | NRF-2020R1A2C2008312 | Doo Hyun Chung Jaemoon Koh Yeon Duk Woo |

The funders had no role in study design, data collection and interpretation, or the decision to submit the work for publication.

### Author contributions

Jaemoon Koh, Yeon Duk Woo, Conceptualization, Validation, Investigation, Writing – review and editing; Hyun Jung Yoo, Jun-Pyo Choi, Sae Hoon Kim, Investigation; Yoon-Seok Chang, Ji Hyung Kim, Yoon Kyung Jeon, Hye Young Kim, Supervision; Kyeong Cheon Jung, Conceptualization, Supervision; Doo Hyun Chung, Conceptualization, Supervision, Writing - original draft

### Author ORCIDs

Jaemoon Koh http://orcid.org/0000-0002-2824-5080
Yeon Duk Woo http://orcid.org/0000-0003-3518-9248
Yoon Kyung Jeon http://orcid.org/0000-0001-8466-9681
Hye Young Kim http://orcid.org/0000-0001-5978-512X
Doo Hyun Chung http://orcid.org/0000-0002-9948-8485

### Ethics

All subjects provided informed written consent to participate in this study. In total 30 healthy control subjects and patients with nonallergic and allergic asthma participated in this study. The Institutional

Review Board of Seoul National University Bundang Hospital approved the human studies (IRB #: B-1901/517-304).

All animal experiments were approved by the Institutional Animal Care and Use Committee at Seoul National University Hospital (SNUH-IACUC). The animals were maintained in an AAALAC International (#001169)-accredited facility in accordance with the Guide for the Care and Use of Laboratory Animals 8th edition.

Reviewer #1 (Public Review): https://doi.org/10.7554/eLife.87536.4.sa1
Reviewer #2 (Public Review): https://doi.org/10.7554/eLife.87536.4.sa2
Author Response https://doi.org/10.7554/eLife.87536.4.sa3

## Additional files

### Supplementary files
• Supplementary file 1. List of reagents used for flow cytometry.
• Supplementary file 2. List of primer sequences used for real-time PCR.
• MDAR checklist

### Data availability
Sequencing data have been deposited in GEO under accession codes GSE205761.

The following dataset was generated:

| Author(s) | Year | Dataset title | Dataset URL | Database and Identifier |
|---|---|---|---|---|
| Koh J, Woo YD | 2022 | De novo fatty acid synthesis in invariant NKT cells is critical for the development of airway hyperresponsiveness | https://www.ncbi.nlm.nih.gov/geo/query/acc.cgi?acc=gse205761 | NCBI Gene Expression Omnibus, GSE205761 |

The following previously published dataset was used:

| Author(s) | Year | Dataset title | Dataset URL | Database and Identifier |
|---|---|---|---|---|
| Oh S, Woo Y, Chung D | 2017 | Genome-wide analysis between murine invariant NKT cells and CD4+ T cells | https://www.ncbi.nlm.nih.gov/geo/query/acc.cgi?acc=GSE103190 | NCBI Gene Expression Omnibus, GSE103190 |

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
