## [Editor Report · eLife assessment]

The study's results offer a **fundamental** insight into how ACC1-mediated fatty-acid synthesis affects the survival and pathogenicity of iNKT cells in allergic asthma. The inclusion of mouse models, involving genetic adjustments and reconstitution experiments, along with the disparities found in iNKT cells between allergic asthma patients and control subjects in human studies, adds **compelling** evidence that substantiates these findings.

---

## [Referee Report · Reviewer #1 (Public Review)]

The manuscript focused on roles of a key fatty-acid synthesis enzyme, acetyl-coA-carboxylase 1 (ACC1), in the metabolism, gene regulation and homeostasis of invariant natural killer T NKT_ cells and impact on these T cells' roles during asthma pathogenesis. The authors presented data showing that the acetyl-coA-carboxylase 1 enzyme regulates the expression of PPARg then the function of NKT cells including the secretion of Th2-type cytokines to impact on asthma pathogenesis. The results are clearcut and data were logically presented.

---

## [Referee Report · Reviewer #2 (Public Review)]

In this study the authors sought to investigate how the metabolic state of iNKT cells impacts their potential pathological role in allergic asthma. The authors used two mouse models, OVA and HDM-induced asthma, and assessed genes in glycolysis, TCA, B-oxidation and FAS. They found that acetyl-coA-carboxylase 1 (ACC1) was highly expressed by lung iNKT cells and that ACC1 deficient mice failed to develop OVA-induced and HDM-induced asthma. Importantly, when they performed bone marrow chimera studies, when mice that lacked iNKT cells were given ACC1 deficient iNKT cells, the mice did not develop asthma, in contrast to mice given wildtype NKT cells. In addition, these observed effects were specific to NKT cells, not classic CD4 T cells. Mechanistically, iNKT cell that lack AAC1 had decreased expression of fatty acid-binding proteins (FABPs) and peroxisome proliferator-activated receptor (PPAR)γ, but increased glycolytic capacity and increased cell death. Moreover, the authors were able to reverse the phenotype with the addition of a PPARg agonist. When the authors examined iNKT cells in patient samples, they observed higher levels of ACC1 and PPARG levels, compared to healthy donors and non-allergic-asthma patients.

---

## [Author Response]

The following is the authors’ response to the previous reviews.

**Reviewer #1 (Public Review):**
The manuscript focused on roles of a key fatty-acid synthesis enzyme, acetyl-coA-carboxylase 1 (ACC1), in the metabolism, gene regulation and homeostasis of invariant natural killer T NKT_ cells and impact on these T cells' roles during asthma pathogenesis. The authors presented data showing that the acetyl-coA-carboxylase 1 enzyme regulates the expression of PPARg then the function of NKT cells including the secretion of Th2-type cytokines to impact on asthma pathogenesis. The results are clearcut and data were logically presented.

Thank you for your input into our work. Your comments have been very helpful in enhancing our work.

**Reviewer #2 (Public Review):**
In this study the authors sought to investigate how the metabolic state of iNKT cells impacts their potential pathological role in allergic asthma. The authors used two mouse models, OVA and HDM-induced asthma, and assessed genes in glycolysis, TCA, B-oxidation and FAS. They found that acetyl-coA-carboxylase 1 (ACC1) was highly expressed by lung iNKT cells and that ACC1 deficient mice failed to develop OVA-induced and HDM-induced asthma. Importantly, when they performed bone marrow chimera studies, when mice that lacked iNKT cells were given ACC1 deficient iNKT cells, the mice did not develop asthma, in contrast to mice given wildtype NKT cells. In addition, these observed effects were specific to NKT cells, not classic CD4 T cells. Mechanistically, iNKT cell that lack AAC1 had decreased expression of fatty acid-binding proteins (FABPs) and peroxisome proliferator-activated receptor (PPAR)γ, but increased glycolytic capacity and increased cell death. Moreover, the authors were able to reverse the phenotype with the addition of a PPARg agonist. When the authors examined iNKT cells in patient samples, they observed higher levels of ACC1 and PPARG levels, compared to healthy donors and non-allergic-asthma patients.

Thank you for your thorough analysis of our work.

**Reviewer #1 (Recommendations For The Authors):**
1. I suggest the authors to remove one copy of the sentence "It should be noted that CD4-CreAcc1fl/fl mice lack ACC expression in both conventional CD4+ T cells and iNKT cells." in Lines 421-423.

We have removed the redundant sentence originally shown in LINES 421-423. Thank you for pointing this out.

**Reviewer #2 (Recommendations For The Authors):**
Overall, this is a very strong study with few concerns.1. Are there tissue specific differences in the iNKT cell populations? The authors examined lung iNKT cells in the Figs 1-3, and used liver NKT cells for the mechanistic studies in Fig 4-5. The studies shown in Fig S2 suggest that ACC1 deficient iNKT cells have developmental defects and impaired homeostatic proliferative capacity. Does ACC1 impact lung and liver iNKT cells similarly and is the lack of allergic asthma in ACC1 deficient iNKT cells due to defective iNKT cell trafficking to the lungs or a failure to survive after transfer (Fig 3)?1. Similarly, are chemokine receptor expression patterns similar between WT and ACC1 deficient iNKTs (Fig 4)?1. The authors data suggest that Tregs are not playing a major role in the regulation of asthma induction in their ACC1 deficient mice, based on FoxP3 expression. Did the authors perform suppressor assays to show that the Tregs function similarly in WT and ACC1 deficient mice?In the revised manuscript, the authors addressed my major concerns.

Thank you for your previous comments. They were very helpful in upgrading our scientific work here.